# Molecular Susceptibility and Treatment Challenges in Melanoma

**DOI:** 10.3390/cells13161383

**Published:** 2024-08-20

**Authors:** Kiran Kumar Kolathur, Radhakanta Nag, Prathvi V Shenoy, Yagya Malik, Sai Manasa Varanasi, Ramcharan Singh Angom, Debabrata Mukhopadhyay

**Affiliations:** 1Department of Pharmaceutical Biotechnology, Manipal College of Pharmaceutical Sciences (MCOPS), Manipal Academy of Higher Education (MAHE), Manipal 576104, Karnataka, India; kirankumar.reddy@manipal.edu; 2Department of Microbiology, College of Basic Science & Humanities, Odisha University of Agriculture & Technology (OUAT), Bhubaneswar 751003, Odisha, India; nagrufus@gmail.com; 3Department of Pharmacy Practice, Manipal College of Pharmaceutical Sciences (MCOPS), Manipal Academy of Higher Education (MAHE), Manipal 576104, Karnataka, India; prathvivs@gmail.com (P.V.S.); yagya.malik1@learner.manipal.edu (Y.M.); 4Department of Biochemistry and Molecular Biology, Mayo Clinic, Jacksonville, FL 32224, USA; saimanasaram@gmail.com (S.M.V.); angom.ramcharan@mayo.edu (R.S.A.)

**Keywords:** melanoma, *BRAF* mutations, *BRAF* inhibitors, *MEK* inhibitors, immunotherapy

## Abstract

Melanoma is the most aggressive subtype of cancer, with a higher propensity to spread compared to most solid tumors. The application of OMICS approaches has revolutionized the field of melanoma research by providing comprehensive insights into the molecular alterations and biological processes underlying melanoma development and progression. This review aims to offer an overview of melanoma biology, covering its transition from primary to malignant melanoma, as well as the key genes and pathways involved in the initiation and progression of this disease. Utilizing online databases, we extensively explored the general expression profile of genes, identified the most frequently altered genes and gene mutations, and examined genetic alterations responsible for drug resistance. Additionally, we studied the mechanisms responsible for immune checkpoint inhibitor resistance in melanoma.

## 1. Introduction

The rising incidence of skin cancer is a severe threat to worldwide public health since it places a heavy demand on labor and the economy [1]. The epidermis and dermis are the two main layers that make up the skin. Melanocytes, keratinocytes, Langerhans cells, and Merkel cells are among the several cell types that make up the epidermis, the outermost layer. Cancer may result from a build-up of anomalies in these cells [1]. Skin cancer has a complicated and diverse etiology impacted by its phenotypic, genetic, and environmental variables. Ultraviolet radiation (UVR) is the most prevalent environmental risk factor. DNA damage caused by UVR may result in oxidative stress, the synthesis of pyrimidine dimers, gene mutations, and other cellular alterations that promote cancer [2]. Melanoma, caused by malfunctioning melanocytes, and non-melanoma skin cancers (NMSC), formed from cells derived from the epidermis, are the two primary forms of skin cancers routinely identified [3]. Globally, NMSC causes around a million new cases and 70,000 deaths, according to GLOBOCAN 2022. The incidence and mortality rates for men are roughly 1.5 times higher. Approximately 95% of skin malignancies are NMSCs or keratinocyte carcinomas. They are divided into two categories: squamous cell carcinoma (SCC) and basal cell carcinoma (BCC). On the other hand, because of its high propensity for metastasis, melanoma—which makes up 1.7% of all occurrences of skin cancer—is the deadliest type [4].

According to studies, UV is the cause of 90% of NMSC and approximately 65% of melanoma. Based on UV exposure or sun damage, the World Health Organization (WHO) 2018 divided melanoma into many kinds (Figure 1 and Table 1) [5]. The updated classification includes various types of melanoma, such as superficial spreading melanoma, nodular melanoma, lentigo maligna melanoma, and acral lentiginous melanoma. These types are distinguished based on their clinical presentation, histological characteristics, and UV-induced damage patterns [6]. Research indicates that UV radiation is responsible for about 90% of non-melanoma skin cancers, which include basal cell carcinoma (BCC) and squamous cell carcinoma (SCC). Long-term UV exposure, particularly from sunlight, is a major risk factor for skin cancer [7]. Superficial spreading melanoma (SSM) is one of the most prevalent forms of melanoma, characterized by its tendency to spread horizontally across the skin’s surface before invading deeper layers. This type of melanoma commonly appears on the trunk and extremities and is closely linked to UV exposure.

Guidelines for staging melanoma have been created by the American Joint Committee on Cancer (AJCC), with scores ranging from 0 (in situ) to IV (metastatic) [8]. Since primary melanoma has a 99% 5-year survival rate and metastatic melanoma has a 27% 5-year survival rate, early diagnosis is essential [9]. A considerable portion of patients (20–30%) receive a diagnosis at the metastatic stage, when the prognosis is noticeably worse [10]. A great deal of research has been performed in this area to learn more about the mechanisms and processes of metastasis and how it progresses [11]. The intricate chain of processes known as invasion, intravasation, circulation, extravasation, and colonization are what causes metastasis. Before going into circulation, malignant cells first invade the tissues around them as circulating tumor cells (CTCs). Disseminated tumor cells (DTCs) are those cells that make it to distant organs and have the ability to develop into secondary cancers [12]. Cancer cells, either as a group or as individual cells, invade the surrounding tissues during the local invasion. An epithelial-to-mesenchymal transition (EMT), which allows cells to pass through the basement membrane, is frequently involved in single-cell invasion. This shift is driven by important transcription factors such as *TWIST*, *SNAIL*, and *ZEB 1* and *2*. Tumor cells enter the stroma after breaching the basement membrane and interact with fibroblasts, endothelial cells, adipocytes, and macrophages, among other stromal cells, to support distinct stages of tumor progression [10].

When tumor cells penetrate endothelial and pericyte barriers to enter microvessels, this process is known as intravasation. Factors such as cyclooxygenase-2 (*COX-2*), metalloproteinase 1 (*MMP1*), metalloproteinase 2 (*MMP2*), vascular endothelial growth factors (*VEGF*), and others help tumors to create new blood vessels (neoangiogenesis). Hemodynamic forces and the body’s immune system create problems for CTCs in the bloodstream. For protection, they group together and communicate with platelets. A CTC’s final destination frequently relies on the body’s blood circulation patterns; it may go into dormancy or start colonizing a new place. When tumor cells enter microvessels, they have three options: they can grow intraluminally, break through the microvessels, and contact organ tissues, or they can enter through spaces between pericytes and endothelial cells. When cancer cells adjust to new surroundings, they can create micrometastases, some of which can grow into healthy secondary tumors [10].

Research has demonstrated that compared to normal skin cells, melanoma cells express and activate more significant quantities of *MMP-2* and *MMP-9*. These enzymes degrade the extracellular matrix, which permits melanoma cells to metastasize—the spread of cancerous cells to other tissues, blood arteries, and lymphatic vessels. Furthermore, by releasing pro-angiogenic proteins and remodeling blood arteries within the tumor, *MMP-2* and *MMP-9* encourage angiogenesis. Weakening immune system components also aids in immunological evasion [13]. Patients with metastatic melanoma are receiving far better care and therapy due to the development of personalized medicine. Patient outcomes have significantly improved with immunotherapy [14]. Even though many patients have negative side effects or do not respond to treatment, precision medicine is essential when determining which treatment plan is best for a given patient. The OMICS-based approach has recently become more popular in the medical field. Researchers hope to thoroughly understand the molecular pathways driving cancer by integrating data from different OMICS-based methodologies [15]. This information may help to create personalized treatment plans and identify new biomarkers for the early identification of melanomas. This review provides an overview of immune checkpoint inhibitors (ICIs) and other therapies in treating melanoma and the genetic and immunological aspects that lead to melanoma resistance.

## 2. Genetics of Melanoma

A melanoma can have many gene alterations, with only a small subset of these mutations being true “drivers” of the tumor—these could be either gain-of-function (GOF)/activating or loss-of-function (LOF)/deleterious mutations that cause continuous proliferation of melanoma cells, leading to uncontrolled tumor growth. Similarly, tumor-suppressor genes are also susceptible to mutations, which, when altered, cease to function, potentially activating downstream growth pathways and enabling unregulated tumor growth [16]. In recent times, various genetic changes, including copy number variants (CNVs) and single-nucleotide variants (SNVs), have emerged. These mutations can be somatic or germline and may lead to either gain-of-function (GOF) or loss-of-function (LOF) [17]. GOF mutations, typically found in oncogenes, frequently influence essential cellular functions like proliferation, growth, metabolism, resistance to apoptosis, and cell cycle regulation, resulting in the abnormal activation of related signaling pathways.

Melanoma has the maximum mutational burden of all cancers due to UV-induced DNA damage and/or errors in DNA replication [9]. In primary melanoma, the most common mutations exist in proto-oncogene B-Raf or v-Raf murine sarcoma viral oncogene homolog B (*BRAF*), in the rat sarcoma gene (*RAS*). Mutations in other genes such as neurofibromin 1 (*NF1*), proto-oncogene receptor tyrosine kinase (*KIT*), telomerase reverse transcriptase (*TERT*) mutations, tumor protein p53 gene (*TP53*), cyclin-dependent kinase inhibitor 2A (*CDKN2A*), cyclin-dependent kinase inhibitor 2B (*CDKN2B*), Phosphatase and tensin homolog (*PTEN*), Ras-related C3 botulinum toxin substrate 1 (*RAC1*), and G Protein Subunit Alpha Q and 11 (*GNAQ/GNA11*), were reported [18]. A summary of the most frequently altered genes in somatic and germline mutations in melanoma and the pathway involved is listed in Table 2 [16,19]. Most frequently altered genes in melanoma subtypes are detailed in Table 3 [20].

According to the geographical distribution, the occurrence of cutaneous melanoma is higher in Caucasians when compared to Hispanic, African American, Indo-American, and Asian population groups [23]. The evaluation of various literature sources found that the following genetic mutations or gene signatures have been found specifically in some geographic regions. *NRAS* and *KIT* gene mutations are associated with the melanoma subtype in the Australian group [24]. There is an association between the prevalence of *BRAF* gene mutations in melanoma patients with respect to different ethnic groups. The highest prevalence is 70% in the USA, and the lowest is 41% in the Russian population [25]. Studies have also shown that the likelihood of developing melanoma in *CDKN2A* carriers varies across geographical areas and increases with age. According to a report by Bishop et al., carriers in Europe had the lowest penetrance of 13% at the age of 50, and the highest penetrance of 91% was in the Australian population at the mean age of 50 [26]. Several studies have also focused on the co-occurrence of the *BRAF* and *NRAS* mutations, suggesting that the concurrent presence of these mutations may complicate treatment options, as they may have distinct responses to targeted therapies [25,27]. The co-occurrence of *BRAF* and *NRAS* mutations in major geographical areas is depicted (Figure 2). A recent study focused on the positional identity linked to the anatomical location as a determining factor for the potential development of melanoma [28]. The *RAS* family of proteins comprises *NRAS* and *BRAF*, both integral in governing cell development, differentiation, and survival processes. *RAS* functions as a molecular switch that becomes activated by extracellular signals, subsequently triggering downstream signaling through *MAP* kinase pathways (Appendix A) [29]. The *MAPK* pathway interfaces multiple transcription factors and is triggered by receptor tyrosine kinases (*RTKs*) to turn on the transcription of several genes. In melanomas, the *MAPK* pathway is activated by mutations in *BRAF* and *NRAS* genes. *BRAF* mutations cause the long-term activation of the *MAPK* pathway, leading to uncontrolled cell growth and proliferation. *BRAF* inhibitors block downstream kinases, causing cell death and suppressing growth and proliferation [30].

All these variants accelerate the transition from primary to metastatic melanoma. In the breakthrough or initial phases, a normal melanocyte develops an initial driver mutation, causing melanocyte hyperplasia and the production of a melanocytic nevus. *BRAF* and *NRAS* mutations are the most common mutations found in melanocyte nevi, with the latter found mostly in congenital nevi [43,44,45]. In the subsequent step, called the expansion phase, certain melanocytic nevi advance into intermediate lesions that develop *TERT* promoter mutations and eventually develop into melanoma in situ [46]. Once the primary melanoma has accumulated several mutations in *CDKN2A*, *TP53*, *PTEN*, and other genes, it enters the invasive phase and transforms into malignant melanoma [45,47]. This is the phase with many genetic variants and a high mutational burden. The expression pattern of the top 20 most frequently altered genes in melanoma was explored using the UALCAN database [48,49] and is presented in Figure 3.

## 3. Understanding the Genetic Changes That Fuel Treatment Resistance: The Melanoma Code

### 3.1. BRAF and MEK Inhibitors

The mutations in the *BRAF* gene significantly impact more than 50% of melanoma cases. A mutation at position *V600E* results in the constant activation of the *BRAF* protein within the *MAPK* pathway [51]. The *BRAF V600* mutation is the most common mutation in melanoma. Two FDA-approved drugs, vemurafenib and dabrafenib, target *BRAF V600* mutations. However, the patient response to these *BRAF* inhibitors is very limited to certain patients. Treatment with *MEK* inhibitor trametinib as a first-line therapy yielded similar results as *BRAF* inhibitors [52]. Patients treated with *BRAF* or *MEK* inhibitors exhibited a fast antitumor response; the drivers of acquired antitumor resistance involve the reactivation of the *MAPK* pathway [52,53]. Additional mutations in the *MAPK* pathway, specifically in the *MEK1* and *MEK2* genes, are one of the most frequent and can reactivate the *MAPK* pathway, enabling the cancer cells to proliferate and advance [54]. Dabrafenib and trametinib were the first *BRAF* and *MEK* inhibitor combination and currently FDA-approved drug combination to treat advanced *BRAFV600*-mutated melanoma patients. The *BRAF* and *MEK* inhibitor combination exhibited better response and survival rates and fewer side effects compared to the standard monotherapy with *BRAF* inhibitors. A total of 15–20% of patients with *BRAFV600E* mutation do not respond to these drugs. Although the *MAPK* pathway is a successful approach for treating *BRAF*-mutated melanoma, the rewiring of signaling pathways inevitably leads to acquired resistance. Tumor cells can adapt to *BRAF-* and *MEK*-inhibitor treatments through dynamic alterations in signaling networks [18,52,55,56].

Hoogstrat et al., in 2015, showed that a *BRAFL505H* mutation in melanoma conferred resistance to vemurafenib treatment. This suggests that the *BRAFL505H* mutation may play a substantial role in therapy resistance [57]. Patients with metastatic *BRAFV600*-mutant melanoma develop resistance to selective *RAF* kinase inhibitors, which results in changes to the *MAPK* pathway that confer resistance. Additionally, the *RAF* inhibitor therapy results in various genetic resistance mechanisms, most notably the reactivation of the *MAPK* pathway [53]. There are several mechanisms by which *BRAF* inhibitors can acquire resistance. Additional mutations in the *MAPK* pathway, specifically in the *MEK1* and *MEK2* genes, are one of the most frequent. Even in the presence of the *BRAF* inhibitor, these alterations can reactivate the pathway, enabling the cancer cells to proliferate and advance [54].

Given the collaborative nature of the *MEK* and *BRAF* genes, drugs that hinder *MEK* proteins can be beneficial in addressing *BRAF* gene alterations present in melanomas. Notably, *MEK* inhibitors such as Trametinib (Mekinist), cobimetinib (Cotellic), and binimetinib (Mektovi) have been developed for this purpose [52,58].

*BRAF* protein reactivation can occur through various mechanisms, including the frequent occurrence of amplified mutated *BRAF* alleles, resulting in elevated *BRAF* protein expression [18]. Consequently, the administered dosage of *BRAF* inhibitors becomes insufficient to inhibit its activity effectively. This overexpression can also induce spontaneous dimerization of the mutated *BRAFV600E* protein, resulting in the revival of the *ERK* signal transduction pathway and subsequent resistance to inhibitors. Additionally, splice variants of *BRAF*, attributed to mutations or epigenetic changes, have been found in a subset of resistant melanomas, such as *p61BRAFV600E*, which forms dimers independently of *RAS* kinase activation, rendering *BRAF* inhibitors ineffective against *BRAFV600E* dimers [59,60,61]. Furthermore, *BRAF* gene amplification has been observed in a proportion of *BRAF* inhibitor resistant tumors, contributing to *ERK* reactivation [62]. Another resistance mechanism involves tumor microheterogeneity, where some cells carry the *BRAF V600* mutation while others remain wild-type for *BRAF*.

### 3.2. Genetic Background of Therapy Resistance

Mitogen-activated protein kinase (*MAPK*) is a signal-transduction pathway involved in various physiological programs, such as cell proliferation, differentiation, development, migration, apoptosis, and transformation. The overall *MAPK* signaling is divided into three families: *MAPK/ERK* (extracellular signal-regulating kinase) family, c-Jun N-terminal kinase (*JNK*), and *p38 MAPK* signaling family [63]. Briefly, in *MAPK-ERK*, signaling occurs downstream of ligand binding to receptor tyrosine kinases (*RTKs*). This binding leads to the activation of signaling adapters *GRB2* by activating *RAS*-GDP (inactive RAS bound GDP) to *RAS*-GTP (active form of *RAS* bound GTP). Activated *RAS*-GTP triggers a cascade of events by activating the protein kinase activity of *RAF* isoforms (*RAF1*, *BRAF*, and *ARAF*). Each *RAF* isoform activates MEK (*BRAF* is the strongest activator). *MEK* phosphorylates and activates *ERK1* and *ERK2*. *ERK* can translocate to the nucleus and phosphorylate several transcription factors, which control the cell cycle progression and cell survival [64,65,66].

In melanoma, a dysregulated *MAPK* pathway can lead to abnormal cell proliferation, survival, invasion, metastasis, and angiogenesis. The *BRAF* (oncogenic driver mutation) is highly mutated in melanomas; approximately 93% of melanomas harbor *BRAF* mutations that have *MAPK* activation in melanoma development [64,65,66]. The *BRAFV600E* mutation is the most frequent mutation (70–88%), which is a GOF mutation with elevated *BRAF* kinase activity and constitutive activation of downstream targets [65,66]. The mutant *KRASQ61*, another common mutation of *KRAS* (Kirsten rat sarcoma viral oncogene homolog) in melanoma, can decrease its intrinsic hydrolytic activity and sustain the active state of *KRAS*. *RAS* overstimulation leads to LOF mutations in *NF1* (Neurofibromin 1). In most melanomas, LOF mutations in *NF1* lose their ability to inactivate *RAS* and promote the stimulation of the *RAF* and its downstream targets, leading to the activation of the *MAPK* pathway and subsequent cell proliferation and survival [66].

In melanoma, several genomic changes have been identified as drivers of acquired resistance to *BRAF* inhibitors. The most prevalent resistance mechanisms encompass *BRAF* splice variants, *BRAF* amplification, neuroblastoma *RAS* viral oncogene homolog (*NRAS*), and mutations in *MEK1/2*. Importantly, these mutations have been linked to distinct disease phenotypes, indicating that the specific genetic alteration can influence the course and characteristics of melanoma in response to treatment [67,68]. Table 4 illustrates the details of *BRAF* inhibitors leading to resistance in melanomas.

Splice variants of the *BRAF* gene play a role in causing resistance by impacting the process of dimerization. In cells containing the normal, or wild-type, *BRAF* gene, activation by *RAS* results in the formation of dimers, either as pairs of identical *BRAF* molecules (*BRAF-BRAF*) or mixed pairs with another protein called *CRAF (BRAF-CRAF)*. Conversely, in cells with *V600E* mutations in the *BRAF* gene, dimerization does not happen, and *MEK* activation occurs through individual, unpaired *BRAF* molecules. Consequently, *BRAF* inhibitors are ineffective in treating melanomas that have the wild-type *BRAF* gene because the dimer pairs, whether they consist of identical or mixed molecules, maintain their signaling capabilities despite the presence of the inhibitors. On the other hand, these inhibitors effectively prevent the activity of single *BRAF* molecules. Interestingly, splice variants of the *BRAFV600E* gene possess the ability to form dimers, allowing them to activate the *MEK* pathway even when *BRAF* inhibitors are administered, thus contributing to resistance mechanisms [67,68].

Alterations in essential components within the *NRAS/BRAF/MEK* pathway result in the revival of previously inhibited *MAPK* pathways or the initiation of alternative signaling routes, such as the phosphatidylinositol 3-kinase (*PI3K*)/protein kinase B (*AKT*) pathway. In certain cases of *BRAF* inhibitor resistance among patients, the activation of *PI3K/AKT* is triggered by the absence of phosphatase and tensin homolog (*PTEN*) expression [69,70]. *BRAF* inhibitor-resistant melanoma cells have an altered genetic makeup that leads to heightened cytoprotective autophagy, allowing uncontrolled tumor cell proliferation and increased melanoma cell invasion due to elevated adenosine triphosphate (ATP) secretion. However, certain mutations function autonomously without affecting downstream pathways, and frequently elevated gene expression in *BRAF* inhibitor-resistant cells is linked to growth factors, their receptors, cell-adhesion molecules, and extracellular matrix interactions, with typical mutations involving receptor tyrosine kinases (*RTKs*) like epidermal growth factor receptor (*EGFR*), platelet-derived growth factor receptor (*PDGFR*), hepatocyte growth factor (*HGF*), or insulin-like growth factor (*IGF*) receptor, thereby instigating parallel signaling pathways [67,68,71]. In cells that become resistant to *BRAF* inhibitors, research has found that multiple receptor tyrosine kinases (*RTKs*) become overactive. This occurs because inhibitors block the normal process of limiting *RTK* activity on the cell surface. As a result, there is an increase in *RTK* levels on the cell surface, which leads to the activation of alternative signaling pathways, contributing to drug resistance [67,68,71,72].

**Table 4 cells-13-01383-t004:** Genetic mutations causing resistance mechanism to *BRAF* inhibitors.

Mutation	Mechanism
*BRAF* gene amplification and splicing	The *BRAF* gene was amplified, which significantly increased the expression (*BRAF* protein) and prompted the reactivation of *ERK* when *BRAF* inhibitors were present. The production of shortened *BRAF* proteins, which contain the kinase domain but lack the *RAS*-binding N-terminus region, can result through alternative splicing and form homodimers that are resistant to *BRAF* inhibitors [59,68,70,73,74,75].
*BRAF* secondary mutations	Patients who were resistant to *BRAF* inhibitors showed secondary mutations in *L505H* or the single-nucleotide alteration *V600E.* The *V600E* mutation raises *BRAF* kinase activity and results in *MEK* inhibitor cross-resistance [57,76].
*MEK1/2* mutations	Without *BRAF* activation, *MEK1/2* mutations could restart downstream *ERK* signaling [73,74,75,77].
Receptor interaction proteins, *RTKs*, or membrane receptors are upregulated	Through the stimulation of parallel pathways or by directly inducing the *RAS* pathway, overexpression or hyperactivation of membrane receptors/*RTKs*, which is partially mediated by *MITF* copy gain, may promote acquired resistance [69,70,78,79,80,81,82,83,84,85,86,87,88,89,90,91,92].
Inconsistencies with in *PI3K-AKT* cascade	*PI3K* and *AKT*-activating mutations enhance *AKT* signaling by promoting anti-apoptotic signals and elevating expression of essential proliferative genes, enabling survival signals independent of *BRAF* [70,93,94,95,96,97,98,99,100,101,102].

In melanoma resistant to *BRAF* inhibitors, overexpressed tyrosine kinase receptors initiate signal transduction from the cell membrane to *MAPK/ERK* kinases, inducing cell division through *ARAF* and *CRAF* kinases rather than *BRAF*, leading to resistance to *BRAF* inhibitor/*MEK* inhibitor treatment as cells with the *BRAFV600E* mutation switch to different *RAF* isoforms (*ARAF* or *CRAF*), ultimately reactivating the *ERK* pathway.

Mutations in the *RAS* gene, a frequently mutated oncogene in human neoplasms, disrupt GTPase activity, maintain the protein in its active GTP-bound state, and mediate signal transduction from tyrosine kinase membrane receptors [18]. Mutations in the *RAS* gene can hyperactivate the *MAPK/ERK* pathway by phosphorylating *ARAF* and *CRAF* proteins, compensating for *BRAF* inhibition and inducing cell division, particularly when *ARAF* or *CRAF* is overexpressed while *BRAF* is inhibited; the mutated *RAS* protein, locked in its permanently activated GTP-bound state, also stimulates *BRAFV600E* dimerization, resulting in *ERK* pathway reactivation and resistance to *BRAF* inhibitors that exclusively target *BRAFV600E* monomers [78,103,104]. Dysregulated signaling through oncogenic *BRAF*, a crucial constituent of the RAS pathway, causes ongoing activation of downstream effectors involved in cell cycle progression, increasing cell division and tumor growth. Moreover, this *RAS* activation promotes enhanced cell survival and heightened resistance to apoptosis [105]. Melanomas can evade cell death mechanisms due to dysregulated signaling in the *MAPK* pathway, leading to increased signaling activity and promoting them to proliferate and survive. For instance, the *MAPK* pathway’s reactivation and activation of the *PI3K/AKT* pathway may lead to the emergence of resistance to *MAPK* inhibitors, which can elevate the expression of the *RAS/RAF/MEK/ERK* network’s genes. However, the activation of other signaling pathways, such as the *PI3K/AKT* pathway, might result in the resistance to *MAPK* inhibitors. Various cytokines and growth factors that activate the *PI3K/AKT* pathway are crucial for regulating cell survival and growth [106]. The *MAPK* pathway can be reactivated and become resistant to *MAPK* inhibitors if the *PI3K/AKT* pathway is engaged, which can promote the production of genes implicated in the process. The MAPK pathway upstream of the *RAS/RAF/MEK/ERK* pathway can be stimulated by *PI3K/AKT* pathway activation. For *RAF*, *MEK*, and *ERK* to be activated, one of these genes, *RAS*, must be present. *RAS* and *RAF* activity can override the *MAPK* pathway’s inhibition by *BRAF* inhibitors and revive it, leading to resistance [107]. An additional 20% of melanomas resistant to *BRAF* inhibitors develop modifications that activate both the *MAPK* cascade and the adaptive *PI3K/AKT* survival pathway, which encompass gain-of-function mutations in *NRAS* and *KRAS*, along with an increased expression of receptor tyrosine kinases such as *EGFR* and *PDGFRß* [108]. Additionally, the induction of angiogenesis further feeds the developing tumor with nutrition and oxygen, thereby aiding in its survival and growth. Furthermore, the increased invasion and metastasis observed in melanoma cells may be caused by dysregulated *RAS* signaling.

Another gene activation that transcribes neurofibromin is the *NF1* gene, a GTPase-activating protein that negatively regulates *RAS*, the initial component of the MAPK signaling pathway, by facilitating the conversion of *RAS-GTP* to *RAS-GDP* through hydrolysis. When neurofibromin is non-functional, it leads to the activation of multiple signaling pathways like *MAPK* and *PI3K*, consequently stimulating cell proliferation and survival [109,110]. Mutations in these proteins can lead to their constant activation, contributing to oncogenic signaling pathways [111]. The epidermal growth factor receptor (*EGFR*) is closely associated with melanoma reoccurrence and progression [112]. Further, by elevating protein synthesis and glucose metabolism, the *PI3K/AKT* pathway can support cancer cells’ survival and development (Figure 4) [113].

The *PTEN* functions as a suppressor gene by regulating the cell cycle through catalyzing *PIP3* dephosphorylation at the 3′ position of the inositol ring, thereby inhibiting the *PI3K/AKT* signaling pathway and, consequently, halting cellular proliferation [116]. The inactivation of the *PTEN* gene is detected in approximately 10% to 35% of melanoma and is a prevalent factor contributing to the resistance to *BRAF* inhibitors [18,117,118]. When *PTEN* protein expression is diminished, it leads to continuous *PI3K/AKT* signaling pathway activation, promoting cell proliferation growth and suppressing apoptosis [118]. The intricacy of the *PI3K/AKT/mTOR* signaling network entails multiple feedback mechanisms, extensive interactions with other signaling pathways, and alternative pathways, creating numerous avenues to bypass the impact of *PI3K* inhibition [119]. The precise mechanism remains incompletely characterized, but recent research has outlined several potential resistance mechanisms, such as the reactivation of *PI3K*, the activation of parallel pathways, and the influence of the tumor microenvironment. The *RAS-RAF-MEK-ERK* pathway exhibits extensive interconnections with *PI3K* signaling. Simultaneous blocking of *PI3K* and mTOR has been observed to trigger a favorable feedback reaction, resulting in the heightened activation of *JAK/STAT*, consequently contributing to the development of drug resistance (Figure 4) [113].

### 3.3. MicroRNA and Melanoma

MicroRNAs (miRNAs) are small, non-coding RNA molecules that regulate gene expression and have emerged as critical players in cancer biology, especially in melanoma. Since their discovery, miRNAs have been found to influence molecular and cellular processes. In melanoma, specific miRNAs regulate key signaling pathways that drive tumor growth, metastasis, and treatment resistance [120]. miRNA profiling studies in melanoma identified a network of over 20 miRNAs dysregulated by aberrant *B-RAF/MKK/ERK* signaling. Similarly, 19 novel miRNA candidates dysregulated in clinical cutaneous melanoma samples [121,122]. For instance, dysregulation of miR-21 was detected in melanoma, with an increased copy number in some melanoma cell lines and upregulated in several highly invasive cell lines [123,124]. miR-21 expression in melanoma leads to the inhibition of *PTEN* and *PDCD4* targets [125,126]. The dysregulation of several miRNAs has been influenced by transcription factors, epigenetic mechanisms, and DNA copy number alterations. In melanomas, few miRNAs act as tumor suppressors or as oncogenes. These miRNAs add complexity to the signaling networks and are seen as a potential therapeutic target for melanoma treatment [126].

## 4. Mechanisms of Resistance to Immune Therapy/Checkpoint Inhibitors

Immune checkpoint inhibitors (ICIs) are drugs that block immune checkpoints such as PD-1 (anti-programmed cell death protein 1), PD-L1 (ligand for PD-1 receptor), and CTLA-4 (cytotoxic T lymphocyte-associated antigen-4), which typically restrain the immune system’s activity. Cancer cells can manipulate the PD-1/PD-L1 pathway to evade immune recognition. *CTLA-4* and *PD-1* are T-cell surface receptors associated with immune suppression and dysfunction. Currently, seven ICIs have been approved by the U.S. FDA, of which one is a *CTLA-4* inhibitor (ipilimumab), three are *PD-1* inhibitors (nivolumab, pembrolizumab, and cemiplimab), and three are *PD-L1* inhibitors (atezolizumab, durvalumab, and avelumab). ICIs have changed the approach of treatment strategy in numerous cancer types. The rapid FDA approval of ICIs can be attributed to clinical trial data demonstrating their superior anti-melanoma effects compared to traditional therapies. Many groups around the world have studied the approach of ICIs in melanomas; most approaches were successful, while in advanced-stage melanoma patients’ application of ICIs, monoclonal antibodies CTLA-4, PD-1, and PD-L1 have produced a good response (a median survival of 5 years). However, most patients show no response to the treatment. Identifying the predictive biomarkers to differentiate between responders and non-responder patients for these therapies is necessary for choosing the treatment regime. The circulating cytokine IL-6 level is a potential biomarker to distinguish between advanced stage and poor prognosis in patients with different cancer types. A recent study suggests that cytokine IL-6 could be the biomarker of disease progression and poor prognosis in *BRAF* wild-type advanced melanoma treated with pembrolizumab. However, more studies are required to establish the definitive role of IL-6 levels as potential prognostic and predictive biomarkers in various groups of melanoma patients [127,128,129,130,131,132,133,134]. Besides the lack of definitive biomarkers, the underlying mechanism for the resistance remains elusive.

The major concern with ICIs is deciphering the intricate resistance mechanisms and to developing novel drug combinations to optimize treatment approaches to overcome the resistance. The resistance mechanism can be primary or acquired. Primary resistance is an inherent lack of response to the treatment; acquired resistance emerges during treatment. An overview of primary and acquired resistance mechanisms to ICI therapy is summarized in Table 5. The mechanism of resistance is categorized as intrinsic or extrinsic to tumor cells. Intrinsic resistances are related to the mechanism specific to the tumor cells involved in immune responses, cell signaling, gene expression, and DNA damage response. Extrinsic resistances are associated externally with the tumor cells throughout the T-cell activation [127].

**Table 5 cells-13-01383-t005:** Mechanisms of resistance against immune checkpoint inhibitors (ICIs).

Drivers	Immune-Evasion Mechanism	References
**Primary resistance**		
*VEGF* and *ANG2* overexpression	TME infiltration by TILs	[135]
*CXCR3*	Restores the cytotoxicity of CD8+ in TME	[136,137]
TMB	Regulates immunotherapy response	[138,139,140,141]
IL-6 and IL-10 levels in TME	Impairs DC maturation	[142,143,144]
**Acquired resistance**		
*B2M*	Regulates MHC class I-mediated tumor antigen presentation in tumor lesions	[127,145]
*JAK1*	Regulates transcription of the IFN- γ-inducible genes and T-cell infiltration	[146]
*EZH2*	i. Reduces the expression of proteins RASSF5 and ITGB2 ii. STING regulation	[147,148]
*KDM5B*	Overexpression of SETDB1 (H3K9 methyl transferase)	[149,150,151,152]
*SETDB1*	Regulates the expression of immune-related gene clusters that encode MHC I antigens	[134,150]
*HDAC6*	Regulates IL-10 and PD-L1 expression	[148]
*FTO*	Regulates PD-1 expression	[153]
*LAG3*	Regulates the activity of T-cells	[154,155,156,157]
*TIM-3*	Regulates the activity of T-cells	[134,158,159]
*SK1*	Regulates S1P, which in turn regulates lymphocyte trafficking and differentiation	[160,161]
*FCRL6*	Regulates cytotoxic NK cells and effector T-cells	[162]
*NLRP3*	Regulates the recruitment into tumor tissues	[163]
Microbiome	Regulates macrophage polarization and DCs activation and CD8+tumor recruitment	[164,165,166,167,168]

VEGF—vascular endothelial growth factor; ANG2—angiopoietin 2; TME—tumor microenvironment; TILs—tumor-infiltrating lymphocytes; CXCR3—C-X-C Motif Chemokine Receptor 3; TMB—Tumor Mutational Burden; DC-dendritic cell; B2M—Beta 2 microglobulin; JAK1—Janus Kinase 1; EZH2—histone methyltransferase; KDM5B—H3K4 demethylase; FTO—m6A RNA demethylase); LAG3—lymphocyte-activation gene 3; TIM-3—T-cell immunoglobulin and mucin domain 3; SK1—Sphingosine kinase 1; S1P—sphingosine-1-phosphate; FCRL6—Fc receptor-like 6 protein; NLRP3—Nucleotide-Binding Domain, Leucine-Rich Containing Family, Pyrin Domain-Containing-3; PMN-MDSCs—granulocytic myeloid-derived suppressor cells.

Transcriptomic analysis from the melanoma patients reveals that responsiveness to the pretreatment with anti-*CTLA-4* showed a positive correlation with increased tumor mutational burden (TMB) and increased expression of neoantigen and cytolytic markers in the immune microenvironment [169]. In the case of anti-*PD-1* pretreatment, responsive melanoma exhibited elevated levels of CD8^+^ T-cell infiltration and expression of *PD-L1* on tumor cells or immune cells; thus, these particular signatures might act as a potential biomarker for treatment responsiveness [170,171,172]. In the melanoma mice model, more infiltration of intratumoral follicular Treg cells reduced responsiveness to anti-*PD1* treatment [173,174]. In melanoma patients, MHC-II expression on tumor cells correlates with a more favorable response to anti-*PD1/PDL1* treatment [175]. In certain individuals, due to immunoediting, the immune system selects subclones of tumor cells lacking expression of neoantigens, causing poor immunogenicity and resistance to ICIs [127]. Altogether, high TMB, increased expression of MHC-II, and depleted levels of Tregs improve the efficacy of anti-*PD1* treatment [173].

### 4.1. Clinical Predictors of Immune Therapy in Metastatic Melanoma

In a multi-institutional retrospective analysis of 229 melanoma patients, 60 patients (26%) had *NRAS G12/G13/Q61* mutations, 53 patients (23%) had *BRAFV600* mutations, and 116 (51%) had neither *NRAS/BRAF* mutations. In response to first-line immune therapy (IL2, ipilimumab, and anti-PD-1/PD-L1), 28% of the *NRAS*-mutant cohort showed a complete response/partial response (CR/PR). In contrast, the *NRAS/BRAF* wild-type cohort exhibited a 16% response (28% vs. 16%, *p* = 0.04), and the best response to any line of immunotherapy was 32% and 20%, respectively (32% vs. 20%, *p* = 0.07). The patients with *NRAS*-mutant melanoma exhibited a heightened response rate and experienced clinical benefit from immune therapy (Table 6). This retrospective study indicates that advanced melanoma with *NRAS* mutations exhibits better immune-based treatment outcomes than non-*NRAS* mutations [176].

We evaluated OS and PFS for patients with *BRAF* mutations from the ICI cohort (Miao_Melanoma-OS and Miao_Melanoma-PFS datasets) to generate survival curves using Kaplan–Meier analysis. We observed an improved OS and PFS trend for patients with *BRAF* mutations (Figure 5).

Besides somatic mutations, CNVs might also aid in the selective response to ICIs. Data from small cohorts of melanoma patients treated with ICIs suggest that the integrity of the IFN-γ pathway is essential for the responsiveness to anti-*PD1* and anti-*CTLA-4* treatment. This indicates that a loss of IFN-γ signaling in tumor cells may promote resistance to immune checkpoint therapies [177,178].

A high mutational load (nonsynonymous mutations per exome) also exhibited a better clinical benefit from ipilimumab treatment. However, the mutational load alone does not effectively indicate *CTLA-4* blockade therapy response. The therapeutic advantages of ipilimumab were observed in correlation with tumor-specific neoantigens. The tumor-specific expression of somatic neoepitopes increased the overall antigenicity trend. Patients with sustained clinical benefits demonstrated the expression of a tetrapeptide neoantigen signature. Similarly, the presence of this tetrapeptide signature correlated strongly with survival. Mutations resulting in the presentation of specific neoepitopes enhance MHC class I binding, eliciting an intensified antitumor response augmented by *CTLA-4* blockade [179].

In a parallel study, a transcriptomic analysis of tumor biopsies from 40 melanoma patients revealed a connection between improved immune therapy response and factors such as a higher mutational load, increased neoantigen load, and elevated expression of cytolytic markers within the immune microenvironment [169]. Single-cell RNA sequencing and computational analyses on 33 melanomas identified a distinct resistance program unique to tumor cells. This program is linked to T-cell exclusion and immune evasion. *CDK4* is one of the key master regulators involved in the resistance program. Counteracting this program through *CDK4/6*-inhibition enhances the responsiveness of melanoma to ICIs in mouse models [180].

### 4.2. Predictive Features of Response to Immune Checkpoint Blockade (ICB)

Auslander et al. developed an immuno-predictive score (IMPRES), a predictor of ICB response in melanoma patients. IMPRES is constructed based on pair-wise relations between the expressions of 28 checkpoint genes with co-stimulatory or co-inhibitory effects. The above study identified seven immune-related consistently differentially expressed pathways (termed CDPs) that are common in all anti-PD-1 datasets and four CDPs common across all anti-CTLA-4 datasets. The correlation between each IMPRES feature and the expression of each of the CDPs was computed. Subsequently, IMPRES was used to predict the response to ICB among melanoma patients. While IMPRES can predict all the true responders, it misses half of the nonresponders. Elevated IMPRES scores correlate with enhanced OS and PFS in melanoma patients treated with ICB [181].

An analysis of copy number variations using whole-exome sequences (WES) from 469 melanoma cases did not identify any specific recurrent variation to either responders or non-responders to immune therapy treatment. *BRAC2* with nsSNVs (6 of 21 tumors) are better responders. These *BRAC2* loss-of-function mutations might lead to a defect in homologous recombination, double-strand DNA break repair, or some unknown effects that add to responsiveness to anti-*PD-1* treatment. Transcriptomic analysis was performed on anti-*PD-1* responding and non-responding tumors to analyze the differentially expressed genes (DEGs). A total of 693 genes were differentially expressed, and relative gene up-expression events were higher in non-responding tumors than in responding tumors. DEGs that are expressed in higher levels in pre-treatment tumors that do not respond encompass genes linked to mesenchymal transition, immunosuppression, chemotaxis of monocytes and macrophages, as well as genes associated with wound healing and angiogenesis. Transcriptomic signatures derived from perturbation-based analysis displayed co-enrichment patterns (9 of 13 non-responding vs. 1 of 15 responding pretreated anti-*PD-1* tumors). These collective signatures are termed as the innate anti-*PD-1* resistance (IPRES) signature. Innately resistant tumors exhibit IPRES, indicating upregulation of events involved in regulating mesenchymal transition, cell adhesion, remodeling of the extracellular matrix (ECM), wound healing, and angiogenesis. Treatment with mitogen-activated protein kinase (*MAPK*) inhibitors causes comparable alterations in residual melanoma. This observation implies that these signatures might negatively impact the responsiveness to anti-*PD-1* therapy [182].

## 5. Gender Differences in Melanoma

Melanoma exhibits notable differences in molecular mechanisms between sexes. These differences can influence disease susceptibility, progression, and response to treatment. Understanding these disparities is crucial for developing tailored prevention and treatment strategies [183]. Recent studies indicate that men and women experience melanoma differently. Males have a higher risk of developing melanoma and a higher risk of mortality compared to female counterparts [184]. In the retrospective study performed on a population of 1023 cutaneous melanoma patients, female melanoma patients showed statistical differences in disease-free survival (DFS) and overall survival (OS) compared to male patients. Men showed a significantly lower median DFS than women (22 vs. 104 months; *p* < 0.001) and OS (20.7 vs. 104 months; *p* < 0.001). Similarly, subgroup analysis revealed a statistically significant difference in DFS and OS favoring females compared to male patients with TNM stages I and II. However, no significant differences were observed in TNM stages III and IV. These data were examined without any consideration given to various therapies undergone by the patients [185]. In general, mortality rates are higher among men than women [186].

A population-based cohort study of 11,774 cutaneous melanoma cases from the Munich Cancer Registry found that females showed a 38% lower risk of death compared to males. Female patients, in general, exhibited thinner tumors and less disease progression and metastasis compared to male patients. These differences might be attributed to differences in tumor–host interaction across genders [184,187,188]. The mechanisms responsible for the gender disparity in melanoma treatment outcomes are not well understood. But several factors, such as hormonal differences, signaling pathways, immune function, oxidative stress response, and gene expression, are likely to contribute to the significant differences in melanoma [189,190]. 

In a retrospective, multicohort analysis of patients with metastatic melanoma, obese male patients showed improved PFS and OS compared to male patients with a normal BMI treated with targeted or immune therapy [191].

## 6. Conclusions

Given its location, melanoma is a reasonably easy malignancy to collect and one that is potentially lethal. The scientific community has learned a great deal about the disease and its progression because of the ease with which patient samples can now be accessed. Melanoma, characterized by its molecular complexity and heterogeneity, poses significant challenges in understanding its pathogenesis and effectively treating it. The disease is driven by genetic mutations, such as those in the *BRAF*, *NRAS*, and *KIT* genes, contributing to its aggressive nature and variability in clinical outcomes [192]. Also, epigenetic alterations, including DNA methylation and histone modifications, play critical roles in melanoma development and therapy resistance [193]. The regulatory impact of microRNAs further complicates the molecular landscape, with certain miRNAs acting as either oncogenes or tumor suppressors, influencing tumor progression, metastasis, and treatment response [121,122,125,126]. We have succeeded in determining the most common mutations in the *BRAF, NRAS*, and *TERT* genes that cause melanoma [194].

Genomic sequencing of cancer patient tumor samples has helped researchers analyze gene abundance and identify treatment strategies. Melanoma has a high mutation burden, with 70–80% of melanomas having *BRAF* mutations. These mutations activate the *MAPK* pathway, which is essential for tumor growth. Survival analysis varies in males and females, with *BRAF* mutations having a higher survival rate than *NRAS*. Patients with *BRAF* mutations have an average of 80 months of OS and 70 months of PFS when using the CAMOIP tool.

The use of OMICS-based approaches has been a boon to researchers in handling large amounts of data, interpreting them, and easily analyzing various mutations, co-factors, and biomarker identification with the latest use of AI in the identification of treatment strategies that benefit a high proportion of patients for the precision targeted therapy. Certain mutations are exclusive to a given region, the co-occurrence of some mutations has been noted, and the survival analysis for the *BRAF* and *NRAS* oncogenes differs significantly in males and females.

Skin cancer is treated using a variety of techniques, such as photodynamic treatment, radiation, cryotherapy, and immunotherapy. An innovative, cost-effective, and effective treatment for skin cancer is needed due to its rising severity and numerous treatment restrictions. Over the last ten years, targeted therapy and immunotherapy have emerged as the two main approaches that have transformed the landscape of systemic treatment for metastatic melanoma.

Several somatic and germline alterations, such as SNVs and CNVs, are linked with melanoma. The oncogenes acquire GOF mutations in the process, as well as LOF mutations, which silence the tumor-suppressor genes. UALCAN is one of the beneficial open-source OMICS instruments used to examine the expression of these tumor-suppressor and oncogene genes. Notably, the finding that *BRAF* mutations are common in about 40–50% of melanoma patients has spurred a renaissance of interest in this sector. This genetic mutation causes the downstream *MAPK* pathway (*RAS/RAF/MEK/ERK* proteins) to be constitutively activated, which is essential for tumor growth. The predominant mutation is *BRAFV600E*, which accounts for over 80% of all *BRAF* alterations. This mutation replaces valine with glutamate through a single nucleotide change (GTG to GAG). Moreover, *V600K* mutations, which involve a two-fold nucleotide change (GTG to AAG) and the substitution of lysine for valine, account for approximately 16% of *BRAF* alterations. These oncogenic alterations significantly influence tumor growth and metastasis initiation and promotion. Novel *BRAF* small-molecule inhibitors, including vemurafenib, dabrafenib, and encorafenib, have been developed as therapeutic approaches for treating melanoma in response to this insight. These inhibitors provide a promising method of treating this difficult condition by focusing on faulty *BRAF* signaling.

ICI treatment has been successful in treating melanoma, but a significant percentage of patients show resistance. Factors influencing resistance include elevated neoantigen, cytolytic markers, PD-L1, and MHC-II expression. New therapeutic approaches should consider the tumor microenvironment and its elements. Advanced melanoma patients with *NRAS* and *BRAF* mutations respond better to treatment. Recent studies have established IMPRES predictors for better treatment outcomes, but finding definitive indicators is challenging due to resistance systems.

The transcriptome analyses of several melanoma patients show that different factors may influence the resistance to immune checkpoint inhibitor treatment, such as elevated expression of neoantigen, cytolytic markers, PD-L1, and MHC-II. New therapeutic approaches could be evaluated with a deeper comprehension of the tumor microenvironment, its elements, and the biochemical makeup and metabolic profile of its constituent cells. A small-cohort, multi-institutional retrospective investigation indicates that advanced melanoma patients with *NRAS* mutations respond better to first-line immunotherapy. *BRAF* mutations from the ICI cohort showed a greater response to the treatment in a similar type of analysis. This suggests that the mutational load, mutations in *NRAS* and *BRAF*, and DEGs of specific targets involved in immunological responses can function as a prognostic marker for the treatment. The methods for predicting characteristics that lead to better treatment outcomes have been established in recent studies. IMPRES is one such predictor; a higher IMPRES score corresponds to increased OS and PFS in melanoma patients receiving ICI treatment. IPRES signatures also suggest the influence of treatment resistance in patients with melanoma. However, it would be difficult to definitively find predictive indicators because of the many systems behind resistance. However, extensive cohorts must be used to validate the discovered predictive markers.

Further, sex-based differences in melanoma susceptibility and treatment responses highlight the need for gender-specific approaches to melanoma management. Variations in hormonal influences, genetic predispositions, and immune responses between sexes can affect disease progression and therapeutic efficacy [183,184]. For example, estrogen has been shown to impact melanoma progression in women, while androgens may influence disease behavior in men. Furthermore, differences in mutation prevalence, such as the higher incidence of *BRAF* mutations in women, underscore the need for personalized treatment strategies [195].

Despite advancements in targeted therapies and immunotherapies, including *BRAF* inhibitors and checkpoint inhibitors, challenges remain in managing resistance and relapse. Resistance mechanisms are multifaceted, involving changes in tumor biology, such as alterations in signaling pathways and immune-evasion strategies [196,197]. The emergence of resistant melanoma clones and the development of secondary mutations necessitate ongoing research to identify novel therapeutic targets and combination strategies.

Future progress in melanoma treatment relies on integrating comprehensive molecular insights into clinical practice. By utilizing advanced genomic, epigenomic, and proteomic analyses, researchers and clinicians can enhance the precision of melanoma treatments. This personalized approach aims to improve patient outcomes through tailored therapies that address the specific molecular characteristics of each patient’s tumor. Continued exploration of the molecular mechanisms underlying melanoma will be crucial in overcoming the existing treatment challenges and achieving better therapeutic success.

New therapeutic approaches could be evaluated with a deeper comprehension of the tumor microenvironment, its elements, and the biochemical makeup and metabolic profile of its constituent cells.

## Figures and Tables

**Figure 1 cells-13-01383-f001:**
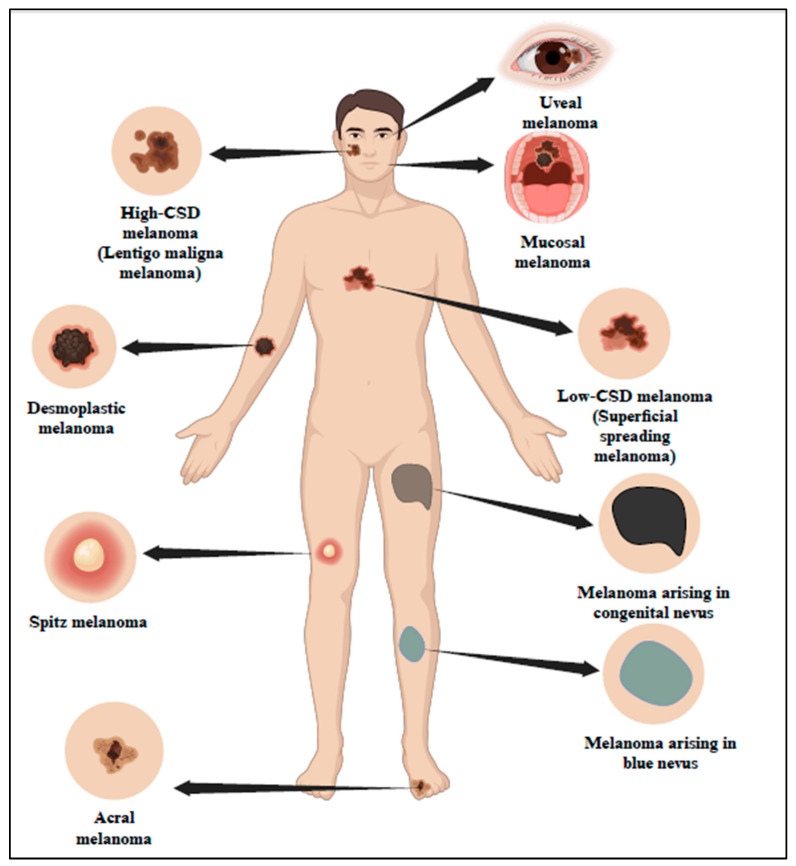
Classification of melanoma in association with sun exposure as per WHO 2018. The 2018 classification by WHO has categorized melanoma into different types based on sun exposure or sun damage. Created with BioRender.com.

**Figure 2 cells-13-01383-f002:**
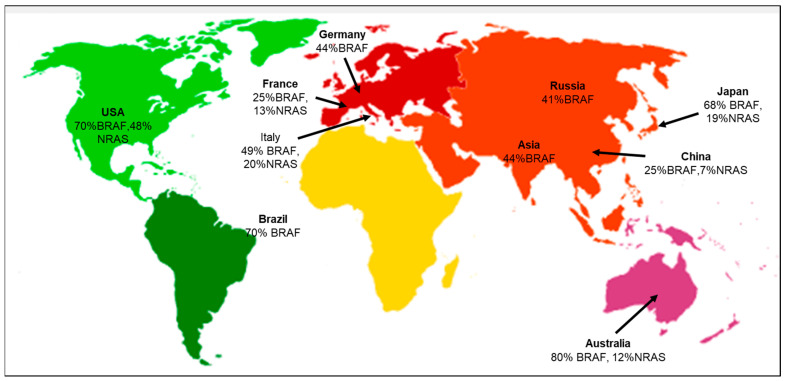
Geographical distribution of co-occurring *BRAF* and *NRAS* mutations [31]. Co-occurrence of both *BRAF* and *NRAS* mutations specific to the various geographical areas, including Germany [32], USA [33,34], Italy [35], France [36], Brazil [37], Asia [38], Russia [39], China [40], Japan [41], and Australia [42].

**Figure 3 cells-13-01383-f003:**
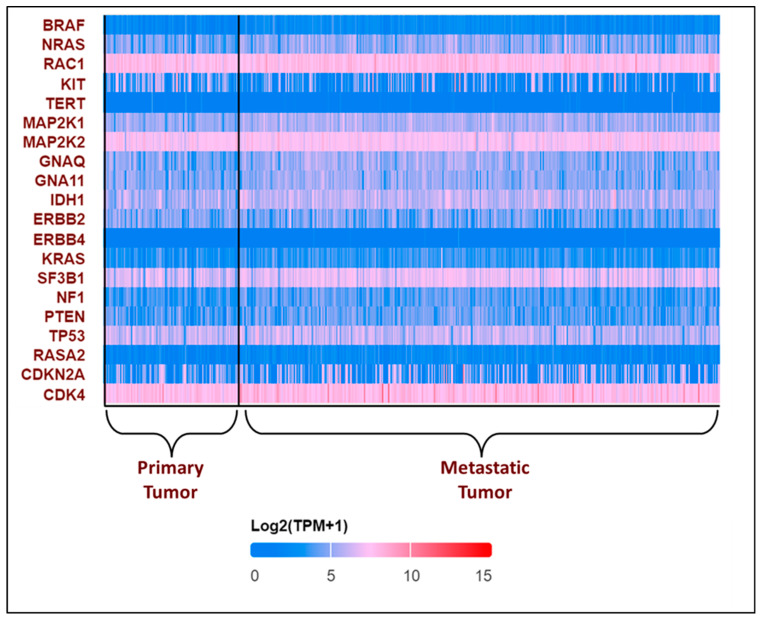
Expression patterns of input genes in skin cutaneous melanoma: The heat map illustrates the expression profiles of various genes in primary and metastatic tumors in melanomas from the TCGA subset. Blue indicates lower expression, and red indicates higher expression on log 2 (TPM + 1) scale using UALCAN. The expression profile of genes associated with melanoma in melanoma patients was obtained from the UALCAN-UAB database [48]. The Y-axis represents the major genes associated with melanoma, and the X-axis represents the tumor type—primary tumor (n = 104) or metastatic tumor (n = 368). The image describes the differential expressions of the primary and the metastatic tumors. The above plot shows that *RAC1*, *MAP2K2*, and *CDK4* consistently show high expression in all the patients, while *TERT* and *ERBB4* show low expression. *BRAF* was found to have a lower expression, with log2(TPM + 1) values lying between 0 and 5, which is not consistent with the existing literature [50]. The expression profile of the other genes varies in each patient.

**Figure 4 cells-13-01383-f004:**
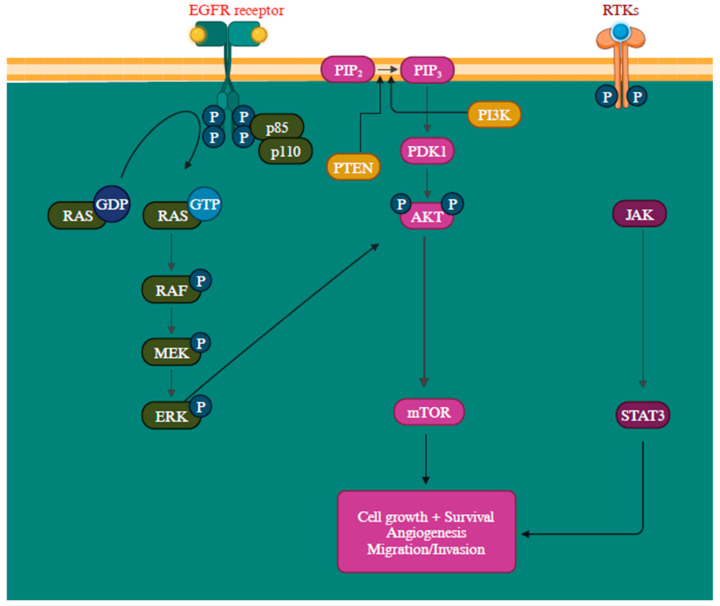
Overview of *PI3K/AKT*, *RAS/MAPK*, and *JAK* (Janus Kinases)/*STAT* (signal transducer and activator of transcription) signaling pathways promoting melanoma metastasis [114,115]. The *MAPK*, *PI3K/AKT*, and *JAK/STAT* pathways are all regulated by the receptor tyrosine kinases (RTKs) [113]. Created with BioRender.com.

**Figure 5 cells-13-01383-f005:**
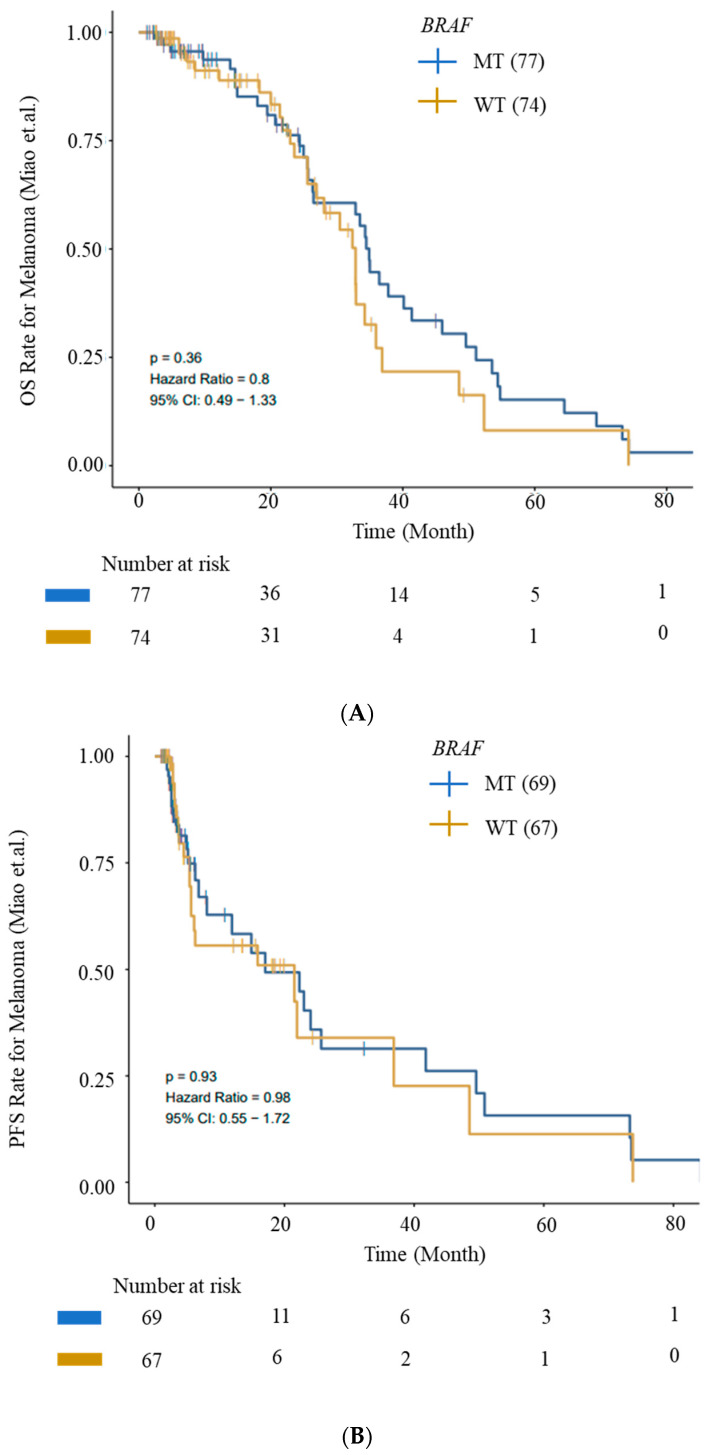
Kaplan–Meier survival curves for OS and PFS. (**A**) OS and (**B**) PFS from ICI Cohort for *BRAF* (wild-type—WT) and *BRAF*-mutant (MT) cohorts. Improved OS and PFS trends for patients with *BRAF* mutations are depicted. Data were generated using the CAMOIP tool.

**Table 1 cells-13-01383-t001:** WHO 2018 classification of melanoma [5].

Melanoma Subtype	Location
**Melanomas arising in sun-exposed skin**	
Low-CSD * melanoma/superficial spreading melanoma	Trunk or extremities
High-CSD * melanoma (including lentigo maligna melanoma and high-CSD * nodular melanoma)	Head and neck region
Desmoplastic melanoma	Head and neck, trunk, or extremities
**Melanomas developing in shielded areas or without known etiological associations with UV radiation exposure**	
Malignant Spitz tumor (Spitz melanoma)	Head and neck, trunk, or extremities
Acral melanoma	Acral sites
Mucosal melanoma	Mucosae
Melanoma arising in congenital nevus	Trunk and proximal parts of the limbs, scalp, or neck
Melanoma arising in blue nevus	Scalp, extremities, or trunk
Uveal melanoma	Eyes

* Cumulative sun damage.

**Table 2 cells-13-01383-t002:** Most frequently altered genes in melanoma [16,19,21,22].

Gene Name	Gene Symbol	Frequency (%)	Most Commonly Reported Mutations	Pathway	Function
	**Somatic GOF/Activating mutations**				
Braf Proto-Oncogene, Serine/Threonine Kinase	*BRAF*	40–60	V600E V600K V600D	MAPK signaling	Cell proliferation and survival
Neuroblastoma RAS Viral Oncogene Homolog	*NRAS*	15–25	G13R G12D Q61H Q61R Q61K Q61L G12S	MAPK/PI3K signaling	Cell proliferation, differentiation, and survival
Ras-related C3 Botulinum Toxin Substrate 1	*RAC1*	~9	P29S P29L P34S P159L V14E E31D	MAPK signaling	Cell proliferation and migration
KIT proto-oncogene receptor tyrosine kinase	*KIT*	10	L576P K642E V559A N822K N822I S451C G226W P36Q	MAPK/PI3K and JAK/STAT signaling	Cell proliferation and survival
Telomerase reverse transcriptase	*TERT*	40–50	−57, T>G, Promoter mutation	Telomerase activity	Cell survival
Mitogen-Activated Protein Kinase 1 and 2	*MAP2K1/2*	~8	K57N E203K F53I E203V Q278H F57V E207K G307C	MAPK signaling	Cell proliferation
G Protein Subunit Alpha Q and 11	*GNAQ/11*	Rare	Q209P R300K K354N R183C H327R S268F R306L P262H R147S R300L	MAPK signaling	Cell proliferation
Isocitrate Dehydrogenase 1	*IDH1*	~8	R132L R132C P33S	Metabolism of isocitrate	Cell proliferation and impaired differentiation
Erb-b2 Receptor Tyrosine Kinase 2/4	*ERBB2/4*	1/19	R138W R190L H809N Q1200H G1056C P551H L403P G573D D150N E969K R196C P943S	Tyrosine kinases signaling	Cell proliferation and survival
Kirsten Rat Sarcoma Viral Oncogene Homolog, GTPase	*KRAS*	1–2	G13D G12D	GTPase activity	Cell proliferation and survival
Splicing Factor 3b Subunit 1	*SF3B1*	10–20	R625C R1297C R625H P228S P465S P370L	Alternative splicing	Tumorigenesis
	**Somatic LOF/deleterious mutations**				
Neurofibromin 1	*NF1*	10–15	Q347 * W1952 * W336 * V341Cfs * 12 Q2239 * R2517 *	MAPK/PI3K signaling	Cell proliferation, differentiation, and survival
Phosphatase and tensin homolog	*PTEN*	4–8	C211 * D19N V217I	PI3K signaling	Apoptosis, cell survival, and immune evasion
Tumor protein p53	*TP53*	15	E294 *	Caspase3, FAS, and CTL-mediated apoptotic pathways	Cell cycle progression, DNA repair, and apoptosis
RAS P21 Protein Activator 2	*RASA2*	~5	R551C	RAS signaling	Cell proliferation and migration
	**Germline LOF/deleterious mutations**				
Cyclin-dependent kinase inhibitor 2A	*CDKN2A*	10–30	Exon 1 deletion Val22Profs * 46 Tyr44 * Trp15 * Ser12 *	RB pathway	Apoptosis and cell survival
Cyclin-dependent kinase 4	*CDK4*	NA	R24C R24H	G1/S phase cell cycle checkpoint	Cell cycle progression

* Indicates protein coding sequence ending at a translation termination codon.

**Table 3 cells-13-01383-t003:** Most frequently altered genes in melanoma subtypes.

Mutation	Frequency (%)
**Cutaneous melanoma**	
*BRAF*	50
*NRAS*	15–20
*KIT*	5
**Acral melanomas**	
*BRAF*	15
*NRAS*	15
*KIT*	15
**Mucosal melanomas**	
*KIT*	15
**Uveal melanomas**	
*GNAQ/GNA11*	>90
**Spitz melanoma**	
*BRAF*	37
*NRAS*	18
*NF1*	11

**Table 6 cells-13-01383-t006:** Overall response rate and clinical benefit to immune therapy [176].

	*NRAS* Mutation	Non-*NRAS* Mutations
Anti-PD-1/PD-L1 (n = 48)		
ORR	64%	30%
CBR	73%	35%
Ipilimumab (n = 169)		
ORR	19%	11%
CBR	42%	19%
first-line immune therapy		
(Kaplan–Meier analysis)—median duration		
PFS	4.1 months	2.9 months
OS	19.5 months	15.2 months

Overall response rate (ORR), CBR; response rate plus stable disease for 24 weeks, overall survival (OS), progression-free survival (PFS), first-line immune therapy (IL2, ipilimumab, and anti-PD-1/PD-L1).

## Data Availability

Data sharing is not applicable.

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
