# Peer review of "Molecular Susceptibility and Treatment Challenges in Melanoma"

_cells, 2024, doi:10.3390/cells13161383_

Round 1
Reviewer 1 Report (New Reviewer)
Comments and Suggestions for Authors
This manuscript overall presents a well-written and comprehensive analysis of contemporary therapeutic strategies for melanoma, examining the topic from multiple perspectives. The authors have highlighted several noteworthy points, particularly regarding the molecular vulnerability and treatment obstacles associated with this disease.
However, the paper could benefit from further refinement in a few areas to strengthen its overall impact.
The introduction provides a general overview of skin cancer, but occasionally feels too rushed and superficial in some details. For instance, the initial sentence in the abstract, "Skin cancer is a prevalent and heterogeneous disease with several subtypes, such as melanoma, basal cell carcinoma, and squamous cell carcinoma," is somewhat basic and doesn't add substantial value to the discussion. This statement is widely known and could either be omitted or replaced with more specific and impactful information that sets the stage for the paper's focus.
Another example of this rushed approach is found in the paragraph: "According to studies, UV is the cause of 90% of NMSC and approximately 65% of melanoma. Based on UV exposure or sun damage, the World Health Organization (WHO) 2018 divided melanoma into many kinds (Figure 1 and Table 1)." This section introduces a lot of data without sufficient depth or explanation. The introduction could be improved by providing more context or elaboration on these points, rather than simply presenting figures.
Regarding Figure 2, the current use of transparent rectangular tables makes the figure somewhat difficult to read. It would be more effective if these tables were presented with a white background instead, ensuring that the information is clear and easily readable.
The review dedicates only a few sentences in the conclusion to potential future directions. A more thorough discussion on how this review could inform and guide future research or clinical practice would strengthen the conclusion and provide a clearer sense of direction for the field.
In conclusion the paper's introduction and conclusion sections would benefit from further development. The introduction should more effectively engage the reader by clearly highlighting the significance of the paper's contributions, particularly the role and impact of molecular classification in melanoma. The conclusion should provide a more concise and powerful summary of the key findings, emphasizing their implications for future research and clinical practice.
Improving these sections will make the manuscript clearer and more effective.
After addressing these points, the paper would be well-suited for consideration for publication.
Author Response
Point-by-point response to Comments and Suggestions for Authors
We thank the reviewers for their positive comments and comments. We have highlighted all the changes as track changes in the manuscript. We hope that the point-by-point response will address the reviewer’s concerns.
Reviewer 1:
Comments 1: This manuscript overall presents a well-written and comprehensive analysis of contemporary therapeutic strategies for melanoma, examining the topic from multiple perspectives. The authors have highlighted several noteworthy points, particularly regarding the molecular vulnerability and treatment obstacles associated with this disease. However, the paper could benefit from further refinement in a few areas to strengthen its overall impact.
Response 1: We thank the reviewers for these comments. We have revised the manuscript according to their suggestions. Please see the highlighted changes made in the manuscript.
Comments 2: The introduction provides a general overview of skin cancer, but occasionally feels too rushed and superficial in some details. For instance, the initial sentence in the abstract, "Skin cancer is a prevalent and heterogeneous disease with several subtypes, such as melanoma, basal cell carcinoma, and squamous cell carcinoma," is somewhat basic and doesn't add substantial value to the discussion. This statement is widely known and could either be omitted or replaced with more specific and impactful information that sets the stage for the paper's focus.
Response 2: We thank the reviewers for this valuable suggestion. We have revised the abstract as per reviewer’s suggestions. Please see highlighted changes in the abstract of the manuscript (Lines 16-17).
Comments 3: Another example of this rushed approach is found in the paragraph: "According to studies, UV is the cause of 90% of NMSC and approximately 65% of melanoma. Based on UV exposure or sun damage, the World Health Organization (WHO) 2018 divided melanoma into many kinds (Figure 1 and Table 1)." This section introduces a lot of data without sufficient depth or explanation. The introduction could be improved by providing more context or elaboration on these points, rather than simply presenting figures.
Response 3: We thank the reviewers for the comments and valuable suggestions. We have revised the manuscript and now expanded this information and tried to present in a detailed manner. Please see the highlighted changes on lines 52- 63.
Comments 4: Regarding Figure 2, the current use of transparent rectangular tables makes the figure somewhat difficult to read. It would be more effective if these tables were presented with a white background instead, ensuring that the information is clear and easily readable.
Response 4: We thank the reviewers for the comments. We have revised manuscript as per reviewer’s suggestions. We have removed the rectangles and kept the background clear.
Comments 5: The review dedicates only a few sentences in the conclusion to potential future directions. A more thorough discussion on how this review could inform and guide future research or clinical practice would strengthen the conclusion and provide a clearer sense of direction for the field.
Response 5: We thank the reviewers for the comments. We have revised the conclusion as per reviewer’s suggestion and highlighted the changes made in the manuscript. Please see Lines 619-628, 630-635, 641-645, 676-683, 702-725.
Comments 6: In conclusion the paper's introduction and conclusion sections would benefit from further development. The introduction should more effectively engage the reader by clearly highlighting the significance of the paper's contributions, particularly the role and impact of molecular classification in melanoma. The conclusion should provide a more concise and powerful summary of the key findings, emphasizing their implications for future research and clinical practice.
Response 6: We thank the reviewers for the comment and suggestions. We have revised manuscript as per reviewer’s suggestions. Please see the highlighted changes made in the manuscript. Page no 21, Line 615 – 723.
Reviewer 2 Report (New Reviewer)
Comments and Suggestions for Authors
In this work Kolathur and coauthors analyze melanoma development and progression looking at key genes and pathways and taking advantage of the new OMICS-based approaches. In addition, they evaluated the impact of innovative therapeutic approaches, in particular immune checkpoint inhibitors (ICIs).
This review is interesting but, given the complexity of the issue, it contains an enormous amount of information, sometimes repetitive, and is not easy for the reader to follow. A suggestion can be to select the main functional genes and/or pathways or, to mention them all, summarize their descriptions. The Conclusion paragraph is once again a summary of the data rather than its interpretation. A general editing is also needed (lines 36, 74, etc).
A major concern is the lack of the important variable represented by sex and/or gender differences. In fact, several recent studies showed differences in melanoma incidence, mortality, response to therapies as well as adverse effects. Different lifestyles also play a role, as women are more interested in sun exposure and tanning, and men are generally less inclined to adopt preventive behaviors. The female survival advantage has been very consistently reported across continents, associated with a lower propensity to metastasize. In addition, many papers have shown that men with advanced melanoma showed a significantly better response to immunotherapy than women, in turn suggesting attention in the use of ICI in women also in view of more serious and frequent adverse effects (see Kudura Cancers, 2022; Conforti 2021, etc). Overall, these disparities suggest that different molecular abnormalities may preferentially underlie melanoma in men and women with a possible different representativeness as biomarkers. The authors should look at sex differences in the molecular mechanisms associated with melanoma. Indeed, without this analysis some relevant information may be lost.
Finally, it could be interesting to cite and possibly introduce a paragraph on miRNAs known to be central players in cancer. Actually, many of them are specifically implicated in the biology of melanoma through modulation of genes involved in the key pathways.
Comments on the Quality of English LanguageSome changes are suggested, not directly related to the English, but rather to the organization of the text, to make it more fluid and easier to follow.
Author Response
|
Point-by-point response to Comments and Suggestions for Authors
Reviwer-2
|
|
Comments 1: In this work Kolathur and coauthors analyze melanoma development and progression looking at key genes and pathways and taking advantage of the new OMICS-based approaches. In addition, they evaluated the impact of innovative therapeutic approaches, in particular immune checkpoint inhibitors (ICIs).
|
|
Response 1: We thank the reviewers for their positive comments.
|
|
Comments 2a: This review is interesting but, given the complexity of the issue, it contains an enormous amount of information, sometimes repetitive, and is not easy for the reader to follow. A suggestion can be to select the main functional genes and/or pathways or, to mention them all, summarize their descriptions. The Conclusion paragraph is once again a summary of the data rather than its interpretation. |
|
Response 2a: We thank the reviewer for this suggestion. We have, accordingly, revised the text and rewrote the conclusions to address the reviewer’s comments. The important genes, mutations and pathways are presented in Table 2. We hope that the revised manuscript will address the reviewer’s concerns. Comments 2b: A general editing is also needed (lines 36, 74, etc). Response 2b: We have revised this sentence please see Lines 36-37 and 87.
Comments 3: A major concern is the lack of the important variable represented by sex and/or gender differences. In fact, several recent studies showed differences in melanoma incidence, mortality, response to therapies as well as adverse effects. Different lifestyles also play a role, as women are more interested in sun exposure and tanning, and men are generally less inclined to adopt preventive behaviors. The female survival advantage has been very consistently reported across continents, associated with a lower propensity to metastasize. In addition, many papers have shown that men with advanced melanoma showed a significantly better response to immunotherapy than women, in turn suggesting attention in the use of ICI in women also in view of more serious and frequent adverse effects (see Kudura Cancers, 2022; Conforti 2021, etc). Overall, these disparities suggest that different molecular abnormalities may preferentially underlie melanoma in men and women with a possible different representativeness as biomarkers. The authors should look at sex differences in the molecular mechanisms associated with melanoma. Indeed, without this analysis some relevant information may be lost.
Response 3: We thank the reviewer for this valuable suggestion. We have revised the manuscript and added a discussion on the genders as suggested by the reviewer. Revised see the new sections in the manuscript. This change can be found – Page number 20, Lines 588 - 612.
Comments 4: Finally, it could be interesting to cite and possibly introduce a paragraph on miRNAs known to be central players in cancer. Actually, many of them are specifically implicated in the biology of melanoma through modulation of genes involved in the key pathways.
Response 4: We appreciate the reviewer’s comment. We have included this information in the revised manuscript. Please see page number 15, line 429-443. Comments 5: Some changes are suggested, not directly related to the English, but rather to the organization of the text, to make it more fluid and easier to follow. Response 5: Thank You for this suggestion. We have reorganized most of the text as per reviewer’s suggestion.
|
This manuscript is a resubmission of an earlier submission. The following is a list of the peer review reports and author responses from that submission.
Round 1
Reviewer 1 Report
Comments and Suggestions for Authors
Comments to the authors
Line 35: The basic characteristic for cancer devlopment is mutations of different genes not the injuries…
Line 44: GLOBOCAN IS ALREADY updeted, authors should use the newest data: https://gco.iarc.fr/en
Line 34: … Keratinocytes, Merkel cells, and Langerhans cells… should not written in capital letters
Lines 129, 513, 567, 584, 619, 639: instead of melanoma tumor authors should use melanoma
Figure 2: the arrows do not point correctly to the countries e.g.: Germany, France…
The source of the map and data are missing, Figure 2 is not cited in the text
Figure 3. Why only those genes (RAC1, MAP2K2, and CDK4) highlighted in the Figure 3 legends? Nothing about one of the most important gene BRAF.
How many primary and how many metastatic tumors were included into the study?
Author Response
We thank the reviewers for the comments and tried to present the data in a detailed manner and the points raised are discussed below.
Reviewer 1
- Line 35: The basic characteristic for cancer development is mutations of different genes, not the injuries…
Response: We thank the reviewer for this comment. We have modified it in the manuscript. Page no. 1, line no. 35-36.
“Any irregularities occurring in this layer, including mutations, can result in several skin injuries, including cancer.”
- Line 44: GLOBOCAN IS ALREADY updated, authors should use the newest data: https://gco.iarc.fr/en
Response: We would like to thank the reviewer for the comment and information. We have updated the information as per the latest GLOBOCAN 2022 reports. Page no. 1, line no. 44-46.
“According to GLOBOCAN 2022, NMSC accounts for over one million new cases and 70,000 deaths globally and has an incidence and mortality that is approximately 1.5 times as high for men as for women.”
- Line 34: … Keratinocytes, Merkel cells, and Langerhans cells… should not written in capital letters
Response: We would like to thank the reviewer for bringing this mistake to our attention. The above suggestion has been implemented. Page no. 1, line no. 34.
- Lines 129, 513, 567, 584, 619, 639: instead of melanoma tumor authors should use melanoma.
Response: We thank the author for this comment. We have replaced melanoma tumor with melanoma.
- Figure 2: the arrows do not point correctly to the countries e.g.: Germany, France…
Response: We would like to sincerely thank the reviewer for bringing this matter to our attention. We have made the necessary corrections to the Figure 2 (Page no. 8)
- The source of the map and data are missing, Figure 2 is not cited in the text.
Response: We would like to sincerely thank the reviewer for bringing this matter to our attention. We have incorporated source in the figure legend (Page no. 8, line no 224-227) and also cited Figure 2 in the text (Page no. 7, line no. 208).
- Figure 3. Why only those genes (RAC1, MAP2K2, and CDK4) highlighted in the Figure 3 legends? Nothing about one of the most important gene BRAF. How many primary and how many metastatic tumors were included into the study?
Response: We would like to thank the author for the comments. We have made the necessary modifications in the legend of Figure 3.
Figure 3. Expression patterns of input genes in skin cutaneous melanoma: The heatmap illustrates the expression profiles of various genes in primary and metastatic tumors in melanomas from the TCGA subset. Blue indicates lower expression and red indicates higher expression on log 2(TPM+1) scale using UALCAN.. The expression profile of genes associated with melanoma in melanoma patients was obtained from the UALCAN-UAB database [39]. The Y-axis represents the major genes associated with melanoma and the X-axis represents the tumor type – Primary tumor (n=104) or Metastatic tumor (n=368). The image describes the differential expressions of the primary and the metastatic tumors. The above plot shows that RAC1, MAP2K2, and CDK4 consistently show high expression in all the patients, while TERT and ERBB4 are showing low expression. BRAF is found to have a lower expression with log2(TPM+1) values lying between 0-5, which is not consistent with the existing literature. The expression profile of the other genes varies in each patient.
Reviewer 2 Report
Comments and Suggestions for Authors
1. This is well written , and address the current field of research but is very lengthy
Concerns :
1. The same Article is available in Preprints .
https://www.preprints.org/manuscript/202309.0120/v1
2. Page 11 - Figure 4 Is is very similar to
Pg 1618 figure 6
Comments on the Quality of English LanguageLine 412 :Melanoma cells can avoid the cell death mechanism owing to the dysregulated signalling through the MAPK pathway, which enables them to endure and develop.
Line 524 For Multi-Institutional retrospective analysis from 229 melanoma patients, 60 pa-524
GTPases can be turned against themselves to encourage invasion and metastasis
Author Response
Reviewer 2
- This is well written, and address the current field of research but is very lengthy
We thank the reviewer for the positive comments.
Concerns:
- The same Article is available in Preprints.
https://www.preprints.org/manuscript/202309.0120/v1
Response: Article mentioned in preprints is the current manuscript that we have submitted to the journal Cells. The Cells journal has published the current submitted article as a preprint to Preprints.org.
- Page 11 - Figure 4 Is is very similar to
https://www.ncbi.nlm.nih.gov/pmc/articles/PMC9485270/pdf/main.pdf Pg 1618 figure 6
Response: We thank reviewer for the comment and to address the reviewer’s concern we have now removed this Figure 4 from the manuscript.
Comments on the Quality of English Language
Line 412 :Melanoma cells can avoid the cell death mechanism owing to the dysregulated signalling through the MAPK pathway, which enables them to endure and develop.
Response: We thank the reviewer for this comment. We have modified it in the manuscript. Page no. 12, line no. 362-364.
“Melanomas can evade cell death mechanisms due to dysregulated signaling in the MAPK pathway, leading to increased signaling activity promoting them to proliferate and survive.”
Line 524 For Multi-Institutional retrospective analysis from 229 melanoma patients, 60 pa-524
Response: We thank the reviewer for this comment. We have modified it in the manuscript. Page no. 16, line no. 461-463.
“In a multi-institutional retrospective analysis of 229 melanoma patients, 60 patients (26%) had NRASG12/G13/Q61 mutations, 53 patients (23%) had BRAFV600 mutations and 116 (51%) had neither NRAS/BRAF mutations.”
GTPases can be turned against themselves to encourage invasion and metastasis.
Response: We thank the reviewer for this comment. We have modified it in the manuscript. Page no. 12, line no. 373-374.
“Mutations in these proteins can lead to their constant activation, contributing to oncogenic signaling pathways”.
We have proofread the manuscript for English.
References:
- Chandrashekar, D.S.; Bashel, B.; Balasubramanya, S.A.H.; Creighton, C.J.; Ponce-Rodriguez, I.; Chakravarthi, B.V.S.K.; Varambally, S. UALCAN: A Portal for Facilitating Tumor Subgroup Gene Expression and Survival Analyses. Neoplasia (New York, N.Y.) 2017, 19, 649–658, doi:10.1016/j.neo.2017.05.002.
Round 2
Reviewer 1 Report
Comments and Suggestions for Authors
The revised manuscript did not really improve. The authors inserted 'Drug Resistance' into the title, but there is nothing about this in the abstract
The revised manuscript did not really improve. The authors inserted 'Drug Resistance' into the title, but they only mention immune checkpoint inhibitors resistance, nothing about the frequently used BRAF inhibitor resistance.
The introduction is still to textbook-like.
line 39-41: Mutations are rather aberrations or alterations, not irregulaties the top layer of skin … of „skin injuries” has different meaning. I highlighted this in review 1. I do not accept the modification.
Line 97: … basal film dissolves … It is not clear what the authors mean on this?
Line 86: What the author wanted to say in this sentence "Numerous studies have been conducted to explore metastasis, to reconcile the knowledge gap on how it develop". Is it referring to discovering the background or the origin of metastasis?
An entire paragraph was deleted (Line 378-449) this was about the BRAF/MEK inhibitors. The authors replaced this part with an overall description of BRAF inhibitors, nothing about the new combinatory inhibitor treatments (dabrafenib/trametinib, vemurafenib/cobimetinib, and encorafenib/binimetinib), only references are cited without any discussion.
BRAFV600 mutation is not consistently written
Paragraph 3.3. Genetic background of therapy resistance: Even a few words were changed the entire paragraph was deleted (similarly to other deleted parts), this way very hard to follow the revised text. Similar problem in Table 4.
BRAFi probably BRAF inhibitor, similarly MEKi is MEK inhibitor: abbreviations are not explained
Figure 2. The arrows do not point correctly Germany and France, and Figure 2 is not cited in the text even I asked for the correction before.
Figure 3. Line 363: “… BRAF is found to have a lower expression with log2(TPM+1) values lying between 0-5, which is not consistent with the existing literature…”. What is the reason for that? This should be explained. References are missing.